# Gradual Domain Adaptation
# without Indexed Intermediate Domains

**Hong-You Chen**
The Ohio State University, USA
chen.9301@osu.edu

**Wei-Lun Chao**
The Ohio State University, USA
chao.209@osu.edu

## Abstract

The effectiveness of unsupervised domain adaptation degrades when there is a large discrepancy between the source and target domains. Gradual domain adaption (GDA) is one promising way to mitigate such an issue, by leveraging additional unlabeled data that gradually shift from the source to the target. Through sequentially adapting the model along the "indexed" intermediate domains, GDA substantially improves the overall adaptation performance. In practice, however, the extra unlabeled data may not be separated into intermediate domains and indexed properly, limiting the applicability of GDA. In this paper, we investigate how to discover the sequence of intermediate domains when it is not already available. Concretely, we propose a coarse-to-fine framework, which starts with a *coarse* domain discovery step via progressive domain discriminator training. This coarse domain sequence then undergoes a *fine* indexing step via a novel cycle-consistency loss, which encourages the next intermediate domain to preserve sufficient discriminative knowledge of the current intermediate domain. The resulting domain sequence can then be used by a GDA algorithm. On benchmark data sets of GDA, we show that our approach, which we name **I**ntermediate **DO**main **L**abeler (**IDOL**), can lead to comparable or even better adaptation performance compared to the pre-defined domain sequence, making GDA more applicable and robust to the quality of domain sequences. Codes are available at https://github.com/hongyouc/IDOL.

## 1   Introduction

The distributions of real-world data change dynamically due to many factors like time, locations, environments, etc. Such a fact poses a great challenge to machine-learned models, which implicitly assume that the test data distribution is covered by the training data distribution. To resolve this generalization problem, unsupervised domain adaption (UDA), which aims to adapt a learned model to the test domain given its unlabeled data [13, 15], has been an active sub-field in machine learning.

Typically, UDA assumes that the "source" domain, in which the model is trained, and the "target" domain, in which the model is deployed, are discrepant but sufficiently related. Concretely, Ben-David et al. [1], Zhao et al. [75] show that the generalization error of UDA is bounded by the discrepancy of the marginal or conditional distributions between domains. Namely, the effectiveness of UDA may degrade along with the increase in domain discrepancy. Take one popular algorithm, self-training [38, 48, 49], for example. Self-training adapts the source model by progressively labeling the unlabeled target data (i.e., pseudo-labels) and using them to fine-tune the model [2, 28, 30, 31, 41, 44, 77, 78]. Self-training works if the pseudo-labels are accurate (as it essentially becomes supervised learning), but is vulnerable if they are not, which occurs when there exists a large domain gap [35].

To address this issue, several recent works investigate *gradual domain adaption (GDA)* [12, 25, 35, 71], in which beyond the source and target domains, the model can access additional unlabeled data from the intermediate domains that shift gradually from the source to the target. By adapting

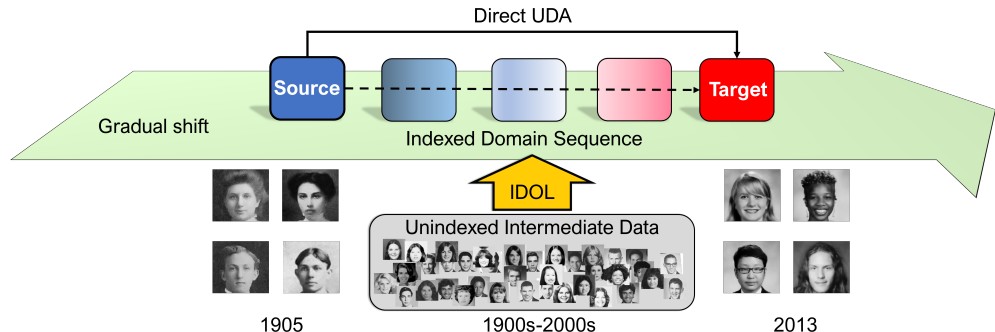

Figure 1: **Gradual domain adaption (GDA) without indexed intermediate domains.** In this setting, one is provided with labeled source, unlabeled target, and additional unlabeled intermediate data that have not been grouped and indexed into a domain sequence. Our approach **intermediate domain labeler (IDOL)** can successfully discover the domain sequence, which can then be leveraged by a GDA algorithm to achieve a higher target accuracy than direct unsupervised domain adaptation (UDA). Images are from the Portraits dataset [14].

the model along the sequence of intermediate domains — i.e., from the ones close to the source to the ones close to the target — the large domain gap between source and target is chipped away by multiple sub-adaptation problems (between consecutive intermediate domains) whose domain gaps are smaller. Namely, every time the model moves a step closer to the target, instead of taking a huge jump that can significantly decrease the performance (see Figure 1 for an illustration). GDA makes sense, as real-world data change gradually more often than abruptly [11, 60, 66]. The recent work by Kumar et al. [35] further demonstrates the strength of GDA both empirically and theoretically.

One potential drawback of GDA is the need for a well-defined sequence of intermediate domains. That is, prior to adaptation, the additional unlabeled data must be grouped into multiple domains and indexed with intermediate domain labels that reflect the underlying data distribution shift from the source to the target. Such information, however, may not be available directly. Existing methods usually leverage side information like time tags [25, 35, 71] to define the sequence, which may be sub-optimal. In some applications, even the side information may not be accessible (e.g., due to privacy concerns), greatly limiting the applicability of GDA.

In this paper, we therefore study GDA in the extreme case — *the additional unlabeled data are neither grouped nor indexed*. Specifically, we investigate how to discover the "domain sequence" from data, such that it can be used to drive GDA. We propose a two-stage coarse-to-fine framework named **I**ntermediate **DO**main **L**abeler (**IDOL**). In the first stage, IDOL labels each intermediate data instance with a *coarse* score that reflects how close it is to the source or target. We study several methods that estimate the distance from a data instance to a domain. We find that a progressively trained domain discriminator — which starts with source vs. target data but gradually adds data close to either of them into training — performs the best, as it better captures the underlying data manifold.

In the second stage, IDOL then builds upon the *coarse* scores to group data into domains by further considering the discriminative (e.g., classification) knowledge the model aims to preserve from the source to the target. Since the additional unlabeled data are fully unlabeled, we take a greedy approach to identify the next intermediate domain (i.e., a group of data) that can best preserve the discriminative knowledge in the current intermediate domain. Concretely, we employ self-training [38] along the domain sequence discovered so far to provide pseudo-labels for the current domain, and propose a novel *cycle-consistency loss* to discover the next domain, such that the model adapted to the next domain can be "adapted back" to the current domain and predict the same pseudo-labels. The output of IDOL is a sequence of intermediate domains that can be used by any GDA algorithms [25, 35, 71].

We validated IDOL on two data sets studied in [35], including Rotated MNIST [36] and Portraits over years [14]. IDOL can successfully discover the domain sequence that leads to comparable GDA performance to using the pre-defined sequence (i.e., by side information). More importantly, IDOL is compatible with pre-defined sequences — by treating them as the coarse sequences — to further improve upon them. We also investigate IDOL in scenarios where the additional unlabeled data may contain outliers and demonstrate IDOL's effectiveness even in such challenging cases. To our knowledge, our work is the first to tackle GDA without grouped and indexed intermediate domains. The success of IDOL opens up broader application scenarios that GDA can contribute to.

## 2 Background: gradual domain adaptation (GDA)

### 2.1 Setup of UDA and GDA

We consider an unsupervised domain adaptation (UDA) problem, in which a classifier is provided with labeled data from the source domain $\mathcal{S} = \{(\boldsymbol{x}_i^{\mathcal{S}}, y_i^{\mathcal{S}})\}_{i=1}^{|\mathcal{S}|}$ and unlabeled data from the target domain $\mathcal{T} = \{\boldsymbol{x}_i^{\mathcal{T}}\}_{i=1}^{|\mathcal{T}|}$. Both $\mathcal{S}$ and $\mathcal{T}$ are sampled IID from the underlying joint distributions of the source $P_{\mathcal{S}}$ and target $P_{\mathcal{T}}$, respectively. A source model $\boldsymbol{\theta}_{\mathcal{S}}$ is typically produced by training on $\mathcal{S}$ directly. The goal of UDA is to learn a target model $\boldsymbol{\theta}_{\mathcal{T}}$ such that it can perform well on the target, measured by a loss $\mathcal{L}(\boldsymbol{\theta}_{\mathcal{T}}, P_{\mathcal{T}})$. We focus on the standard UDA setup that assumes no label shifts [6].

**Gradual domain adaptation (GDA)**, on top of UDA, assumes the existence of a sequence of domains $P_0, P_1, \ldots P_{M-1}, P_M$, where $P_0$ and $P_M$ are the source domain and target domain, respectively, along with $M-1$ intermediate domains $P_1, \ldots, P_{M-1}$. The data distributions of domains are assumed to change gradually from the source to the target along the sequence. One can access unlabeled data $\mathcal{U}_m = \{\boldsymbol{x}_i^{\mathcal{U}_m}\}_{i=1}^{|\mathcal{U}_m|}$ from each of the intermediate domain, forming a sequence of unlabeled data $\mathcal{U}_1, \ldots, \mathcal{U}_{M-1}$. We denote the union of them by $\mathcal{U} = \{\boldsymbol{x}_i^{\mathcal{U}}\}_{i=1}^{|\mathcal{U}|}$. For brevity, we assume that each $\mathcal{U}_m$ is disjoint from the others but the size $|\mathcal{U}_m|$ is the same $\forall m \in \{1, \cdots, M-1\}$.

### 2.2 GDA with pre-defined domain sequences using self-training [35]

We summarize the theoretical analysis of GDA using self-training by Kumar et al. [35] as follows.

**Self-training for UDA.** Self-training [38] takes a pre-trained (source) model $\boldsymbol{\theta}$ and the target unlabeled data $\mathcal{T}$ as input. Denote by $\boldsymbol{z} = f(\boldsymbol{x}; \boldsymbol{\theta})$ the predicted logits $\boldsymbol{z}$, self-training first applies $f(\cdot; \boldsymbol{\theta})$ to target data points $\boldsymbol{x} \in \mathcal{T}$, sharpens the predictions by `argmax` into pseudo-labels, and then updates the current model $\boldsymbol{\theta}$ by minimizing the loss $\ell$ (e.g., cross entropy) w.r.t the pseudo-labels,

$$\boldsymbol{\theta}_{\mathcal{T}} = \texttt{ST}(\boldsymbol{\theta}, \mathcal{T}) = \arg\min_{\boldsymbol{\theta}' \in \boldsymbol{\Theta}} \frac{1}{|\mathcal{T}|} \sum_{i=1}^{|\mathcal{T}|} \ell\big(f(\boldsymbol{x}_i; \boldsymbol{\theta}'), \texttt{sharpen}(f(\boldsymbol{x}_i; \boldsymbol{\theta}))\big), \tag{1}$$

where $\boldsymbol{\theta}_{\mathcal{T}}$ denotes the resulting target model and $\texttt{ST}(\boldsymbol{\theta}, \mathcal{T})$ denotes the self-training process. A baseline use of self-training for UDA is to set $\boldsymbol{\theta}$ as the source model $\boldsymbol{\theta}_{\mathcal{S}}$. However, when the domain gap between $\mathcal{S}$ and $\mathcal{T}$ is large, $\boldsymbol{\theta}_{\mathcal{S}}$ cannot produce accurate pseudo-labels for effective self-training.

**Gradual self-training for GDA.** In the GDA setup, self-training is performed along the domain sequence, from the current model $\boldsymbol{\theta}_m$ (updated using $\mathcal{U}_m$ already) to the next one $\boldsymbol{\theta}_{m+1}$ using $\mathcal{U}_{m+1}$

$$\boldsymbol{\theta}_{m+1} = \texttt{ST}(\boldsymbol{\theta}_m, \mathcal{U}_{m+1}), \tag{2}$$

starting from the source model $\boldsymbol{\theta}_0 = \boldsymbol{\theta}_{\mathcal{S}}$. By doing so, the model gradually adapts from $\mathcal{S}$ to $\mathcal{T}$ through the sequence $\mathcal{U}_1, \ldots, \mathcal{U}_M$ to produce the final target model $\boldsymbol{\theta}_{\mathcal{T}}$.

$$\boldsymbol{\theta}_{\mathcal{T}} = \boldsymbol{\theta}_M = \texttt{ST}\big(\boldsymbol{\theta}_{\mathcal{S}}, (\mathcal{U}_1, \ldots, \mathcal{U}_M)\big). \tag{3}$$

As the domain gap between consecutive domains is smaller than between $\mathcal{S}$ and $\mathcal{T}$, the pseudo-labels will be more accurate in each step. As a result, self-training in GDA can lead to a lower target error than in UDA. We note that the domain sequences in [35] are pre-defined using side information.

**Theoretical analysis.** Kumar et al. [35] make three assumptions for theoretical analysis:

- Separation: for every domain $P_m$, the data are separable such that there exists an $R$-bounded (linear) classifier $\boldsymbol{\theta}_m \in \boldsymbol{\Theta}$ with $\|\boldsymbol{\theta}_m\|_2 \leq R$ that achieves a low loss of $\mathcal{L}(\boldsymbol{\theta}_m, P_m)$.
- Gradual shift: for the domain sequence $P_0, \ldots, P_m$ with no label shifts, the maximum per-class Wasserstein-infinity distance $\rho(P_m, P_{m+1})$ between consecutive domains should be $\leq \rho < \frac{1}{R}$.
- Bounded data: finite samples $X$ from each domain $P_m$ are bounded, i.e., $E_{X \sim P_m}[\|X\|_2^2] \leq B^2$.

Under these assumptions of GDA, if the source model $\theta_0$ has a low loss $\mathcal{L}(\theta_0, P_0)$ on $P_0$, then with a probability $\delta$, the resulting target model $\theta_M$ through gradual self-training has

$$\mathcal{L}(\boldsymbol{\theta}_M, P_M) \leq \beta^{M+1}\Big(\mathcal{L}(\boldsymbol{\theta}_0, P_0) + \frac{4BR + \sqrt{2\log 2M/\delta}}{\sqrt{|\mathcal{U}|}}\Big), \quad \text{where } \beta = \frac{2}{1 - \rho R}. \tag{4}$$

That is, the target error is controlled by the intermediate domain shift $\rho$. If we can sample infinite data (i.e., $|\mathcal{U}| \to \infty$) and the source classifier is perfect (i.e., $\mathcal{L}(\theta_0, P_0) = 0$), with a small distance $\rho(P_m, P_{m+1}) \leq \rho$ between consecutive domains, gradual self-training achieves a zero target error.

# 3 GDA without indexed intermediate domains

## 3.1 Setup and motivation

In this paper, we study the case of GDA where the additional unlabeled data $\mathcal{U}$ are not readily divided into domain sequences $\mathcal{U}_1, \cdots, \mathcal{U}_{M-1}$. In other words, we are provided with a single, gigantic unlabled set that has a wide range of support from the source to the target domain. One naive way to leverage it is to adapt from $\mathcal{S}$ to $\mathcal{U}$[1] and then to $\mathcal{T}$, which however leads to a much larger target error than applying self-training along the properly indexed intermediate domains [35]. We attribute this to the large $\rho(\mathcal{S}, \mathcal{U})$ or $\rho(\mathcal{U}, \mathcal{T})$ according to Equation 4. To take advantage of $\mathcal{U}$, we must separate it into intermediate domains such that $\rho(P_m, P_{m+1})$ is small enough for every $m \in \{0, \dots, M-1\}$.

## 3.2 Overview of our approach: intermediate domain labeler (IDOL)

Given the source data $\mathcal{S}$, the target data $\mathcal{U}_M = \mathcal{T}$, and the unlabeled intermediate data $\mathcal{U} = \{x_i^{\mathcal{U}}\}_{i=1}^{|\mathcal{U}|}$, we propose the intermediate domain labeler (IDOL), whose goal is to sort intermediate data instances in $\mathcal{U}$ and chunk them into $M-1$ domains $\mathcal{U}_1, \cdots, \mathcal{U}_{M-1}$ (with equal sizes) for GDA to succeed.

Taking gradual self-training in Equation 3 as an example, IDOL aims to solve the following problem

$$\min \mathcal{L}(\boldsymbol{\theta}_{\mathcal{T}}, P_{\mathcal{T}}),$$
$$\text{s.t. } \boldsymbol{\theta}_{\mathcal{T}} = \mathtt{ST}\big(\boldsymbol{\theta}_{\mathcal{S}}, (\mathcal{U}_1, \dots, \mathcal{U}_M)\big) \quad \text{and} \quad (\mathcal{U}_1, \dots, \mathcal{U}_{M-1}) = \mathrm{IDOL}(\mathcal{S}, \mathcal{T}, \mathcal{U}; M-1), \quad (5)$$

where the target model $\boldsymbol{\theta}_{\mathcal{T}}$ is produced by applying gradual self-training to the domain sequence discovered by IDOL. IDOL accepts the number of intermediate domains $M-1$ as a hyper-parameter.

Solving Equation 5 is hard, as evaluating any output sequence needs running through the entire gradual self-training process, not to mention that we have no labeled target data to estimate $\mathcal{L}(\boldsymbol{\theta}_{\mathcal{T}}, P_{\mathcal{T}})$. In this paper, we propose to solve Equation 5 approximately via a coarse-to-fine procedure.

**The coarse stage.** We aim to give each $x_i^{\mathcal{U}} \in \mathcal{U}$ a score $q_i$, such that it tells $x$'s position in between the source and target. Specifically, a higher/lower $q_i$ indicates that $x_i^{\mathcal{U}}$ is closer to the source/target domain. With these scores, we can already obtain a coarse domain sequence by *sorting them in the descending order and dividing them into $M-1$ chunks*.

**The fine stage.** Instead of creating the domain sequence $\mathcal{U}_1, \cdots, \mathcal{U}_{M-1}$ by looking at the score of each individual $x_i^{\mathcal{U}} \in \mathcal{U}$, we further consider how data grouped into the same domain can collectively preserve the discriminative knowledge from the previous domains — after all, the goal of UDA or GDA is to pass the discriminative knowledge from the labeled source domain to the target domain. To this end, we propose a novel cycle-consistency loss to refine the coarse scores progressively.

## 3.3 The coarse stage: assigning domain scores

In this stage, we assign each $x_i^{\mathcal{U}} \in \mathcal{U}$ a score $q_i$, such that higher/lower $q_i$ indicates that $x_i^{\mathcal{U}}$ is closer to the source/target domain. We discuss some design choices. For brevity, we omit the superscript $^{\mathcal{U}}$.

**Confidence scores by the classifier $f(x; \boldsymbol{\theta})$.** We first investigate scoring each intermediate data instance by the confidence score of the source model $\boldsymbol{\theta}_{\mathcal{S}}$, i.e., $q_i = \max f(x_i; \boldsymbol{\theta}_{\mathcal{S}})$ across classes. This is inspired by outlier detection [42] and the common practice of self-training, which selects only data with sufficiently large confidence scores to fine-tune upon [35, 38]. Here we employ an advanced version, which is to use the latest $\boldsymbol{\theta}$ (i.e., the model adapted to the current intermediate domain) to select the next intermediate domain. That is, every time we select the highest confidence instances from the *remaining* ones in $\mathcal{U}$ to adapt $\boldsymbol{\theta}$. Overall, we can give data selected in the $m$-th round ($m \in \{1, \cdots, M-1\}$) a score $\frac{M-1-m}{M-2} \in [0, 1]$. One drawback of this method is the blindness to the target domain $\mathcal{T}$. Specifically, we find that using confidence scores tends to select easily classified examples, which do not necessarily follow the gradual shift from the source to the target.

**Manifold distance with the source features.** To model the flow from the source to the target, we argue that it is crucial to consider both sides as references. We thus extract features of $\mathcal{S}$, $\mathcal{T}$, and $\mathcal{U}$ using the source model $\boldsymbol{\theta}_{\mathcal{S}}$ and apply a manifold learning algorithm (e.g., UMAP [50]) to discover the

---

[1]In [35], the authors investigated sampling $\mathcal{U}_1, \cdots, \mathcal{U}_{M-1}$ from $\mathcal{U}$ uniformly at random with replacement.

underlying manifold. Denote by $\gamma(\boldsymbol{x})$ the dimension-reduced features of $\boldsymbol{x}$ after UMAP, we compute the ratio of its distance to the nearest point in the $\mathcal{T}$ and $\mathcal{S}$ as the score $q_i = \frac{\min_{\boldsymbol{x}^{\mathcal{T}} \in \mathcal{T}} \|\gamma(\boldsymbol{x}_i) - \gamma(\boldsymbol{x}^{\mathcal{T}})\|_2}{\min_{\boldsymbol{x}^{\mathcal{S}} \in \mathcal{S}} \|\gamma(\boldsymbol{x}_i) - \gamma(\boldsymbol{x}^{\mathcal{S}})\|_2}$.

**Domain discriminator.** We investigate another idea to emphasize the roles of the source and target, which is to train a domain discriminator [13]. Concretely, we construct a binary classifier $g(\cdot; \boldsymbol{\phi})$ using deep neural networks, which is trained to separate the source data $\mathcal{S}$ (class: 1) and the target data $\mathcal{T}$ (class: 0). We use a binary-cross entropy loss to optimize the learnable parameter $\boldsymbol{\phi}$

$$\mathcal{L}(\boldsymbol{\phi}) = -\frac{1}{|\mathcal{S}|} \sum_{\boldsymbol{x}^{\mathcal{S}} \in \mathcal{S}} \log(\sigma(g(\boldsymbol{x}^{\mathcal{S}}; \boldsymbol{\phi}))) - \frac{1}{|\mathcal{T}|} \sum_{\boldsymbol{x}^{\mathcal{T}} \in \mathcal{T}} \log(1 - \sigma(g(\boldsymbol{x}^{\mathcal{T}}; \boldsymbol{\phi}))), \tag{6}$$

where $\sigma$ is the sigmoid function. We can then assign a score to $\boldsymbol{x}_i \in \mathcal{U}$ by $q_i = g(\boldsymbol{x}_i; \boldsymbol{\phi})$.

**Progressive training for the domain discriminator.** The domain discriminator $g(\cdot; \boldsymbol{\phi})$, compared to the manifold distance, better contrasts the source and target domains but does not leverage any of the data $\boldsymbol{x} \in \mathcal{U}$. That is, it might give a faithful score to an example $\boldsymbol{x}$ that is very close to $\mathcal{S}$ or $\mathcal{T}$, but for other examples that are far away and hence out-of-domain from both ends, the score by $g(\cdot; \boldsymbol{\phi})$ becomes less reliable, not to mention that neural networks are usually poorly calibrated for these examples [21, 42]. We therefore propose to progressively augment the source and target data with $\boldsymbol{x} \in \mathcal{U}$ that has either a fairly high or low $g(\boldsymbol{x}; \boldsymbol{\phi})$, and fine-tune the domain discriminator $g(\boldsymbol{x}; \boldsymbol{\phi})$. We perform this process for $K$ rounds, and every time include $\frac{|\mathcal{U}|}{2K}$ new examples into the source and target. By doing so, the domain discriminator $g(\boldsymbol{x}; \boldsymbol{\phi})$ is updated with data from $\mathcal{U}$ and thus can provide more accurate scores to distinguish the remaining examples into the source or target side. More specifically, let $N$ denote $|\mathcal{U}|$, we repeat the following steps for $k$ from 1 to $K$:

1. Train $g(\cdot, \boldsymbol{\phi})$ using $\mathcal{S}$ and $\mathcal{T}$, based on the loss in Equation 6.
2. Predict $\hat{q}_i = g(\boldsymbol{x}_i, \boldsymbol{\phi}), \forall \boldsymbol{x}_i \in \mathcal{U}$.
3. Rank all $\hat{q}_i$ in the descending order.
4. The data with the $\frac{N}{2K}$ largest $\hat{q}_i$ scores form a new intermediate domain for the source side (their $q_i$ is set to $\frac{2K-k}{2K}$). We remove them from $\mathcal{U}$ and add them into $\mathcal{S}$.
5. The data with the $\frac{N}{2K}$ smallest $\hat{q}_i$ scores form a new intermediate domain for the target side (their $q_i$ is set to $\frac{k}{2K}$). We remove them from $\mathcal{U}$ and add them into $\mathcal{T}$.

Overall, we give a data instance $\boldsymbol{x} \in \mathcal{U}$ selected in the $k$-th round ($k \in \{1, \cdots, K\}$) a score $\frac{2K-k}{2K}$ if it is added into the source side, or $\frac{k}{2K}$ if it is added into the target side.

### 3.4 The fine stage: cycle-consistency for refinement

The domain scores developed in subsection 3.3 can already give each individual example $\boldsymbol{x} \in \mathcal{U}$ a rough position of which intermediate domain it belongs to. However, for GDA to succeed, we must consider intermediate data in a collective manner. That is, each intermediate domain should preserve sufficient discriminative (e.g., classification) knowledge from the previous one, such that after $M$ rounds of adaptation through GDA, the resulting $\boldsymbol{\theta}_{\mathcal{T}}$ in Equation 5 will be able to perform well on $\mathcal{T}$. This is reminiscent of some recent claims in UDA [46, 75]: matching only the marginal distributions between domains (i.e., blind to the class labels) may lead to sub-optimal adaptation results.

*But how can we measure the amount of discriminative knowledge preserved by an intermediate domain (or in the target domain) if we do not have its labeled data?* To seek for the answer, we revisit Equation 4 and consider the case $M = 1$. If the two domains $P_0$ and $P_1$ are sufficiently close and the initial model $\boldsymbol{\theta}_0$ is well-trained, the resulting model $\boldsymbol{\theta}_1$ after self-training will perform well. Reversely, we can treat such a well-performing $\boldsymbol{\theta}_1$ as the well-trained initial model, and apply self-training again — this time from $P_1$ back to $P_0$. The resulting model $\boldsymbol{\theta}_0'$, according to the same rationale, should perform well on $P_0$. In other words, $\boldsymbol{\theta}_0'$ and $\boldsymbol{\theta}_0$ should predict similarly on data sampled from $P_0$. The **similarity between $\boldsymbol{\theta}_0'$ and $\boldsymbol{\theta}_0$ in terms of their predictions** therefore can be used as a proxy to measure **how much discriminative knowledge $\boldsymbol{\theta}_1$, after adapted to $P_1$, preserves**. It is worth noting that measuring the similarity between $\boldsymbol{\theta}_0'$ and $\boldsymbol{\theta}_0$ requires no labeled data from $P_0$ or $P_1$.

Now let us consider a more general case that $M$ is not limited to 1. Let us set $\boldsymbol{\theta}_0 = \boldsymbol{\theta}_{\mathcal{S}}$, where $\boldsymbol{\theta}_{\mathcal{S}}$ has been trained with the labeled source data $\mathcal{S}$ till convergence. Based on the concept mentioned

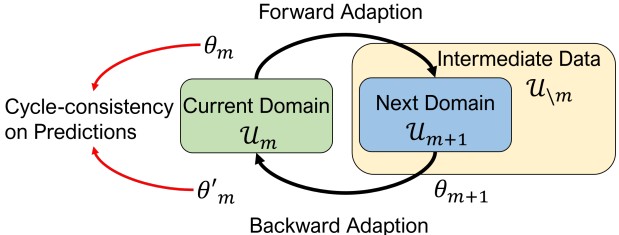

Figure 2: Cycle-consistency (cf. Equation 8).

above, we propose a novel learning framework to discover the domain sequences,

$$\arg\min_{(\mathcal{U}_1,\ldots,\mathcal{U}_{M-1})}\mathbb{E}_{\boldsymbol{x}\sim P_0}\left[\ell\big(f(\boldsymbol{x};\boldsymbol{\theta}_0'),\texttt{sharpen}(f(\boldsymbol{x};\boldsymbol{\theta}_0))\big)\right],$$
$$\text{s.t., } \boldsymbol{\theta}_0' = \texttt{ST}(\boldsymbol{\theta}_M,(\mathcal{U}_{M-1},\ldots,\mathcal{U}_0)), \tag{7}$$
$$\boldsymbol{\theta}_M = \texttt{ST}(\boldsymbol{\theta}_0,(\mathcal{U}_1,\ldots,\mathcal{U}_M)),$$

where $\mathcal{U}_0$ is the source data $\mathcal{S}$ with labels removed, and $\mathcal{U}_M = \mathcal{T}$ is the target data. $\boldsymbol{\theta}_0'$ is a gradually self-trained model from $\boldsymbol{\theta}_0$ that undergoes a cycle $\mathcal{U}_1 \to, \cdots, \to \mathcal{U}_M \to, \cdots, \to \mathcal{U}_0$. In other words, Equation 7 aims to find the intermediate domain sequences such that performing gradual self-training along it in a forward and then backward fashion leads to cycle-consistency [76].

**Optimization.** Solving Equation 7 is still not trivial due to its combinatorial nature. We therefore propose to solve it **greedily, starting with discovering $\mathcal{U}_1$, and then $\mathcal{U}_2$, and then so on.** Concretely, we decompose Equation 7 into a series of sub-problems, as illustrated in Figure 2, each is defined as

$$\arg\min_{\mathcal{U}_{m+1}\subset\mathcal{U}_{\backslash m}} \quad \frac{1}{|\mathcal{U}_m|}\sum_{\boldsymbol{x}\in\mathcal{U}_m}\ell\big(f(\boldsymbol{x};\boldsymbol{\theta}_m'),\texttt{sharpen}(f(\boldsymbol{x};\boldsymbol{\theta}_m))\big),$$
$$\text{s.t., } \quad \boldsymbol{\theta}_m' = \texttt{ST}(\boldsymbol{\theta}_{m+1},\mathcal{U}_m), \tag{8}$$
$$\boldsymbol{\theta}_{m+1} = \texttt{ST}(\boldsymbol{\theta}_m,\mathcal{U}_{m+1}),$$

where $\boldsymbol{\theta}_m$ is the model already adapted to the current domain $\mathcal{U}_m$, and $\mathcal{U}_{\backslash m} = \mathcal{U} \setminus \cup_{j=1}^m \mathcal{U}_j$ is the remaining data that have not been selected by the $m$ intermediate domains so far. That is, each sub-problem discovers only the next intermediate domain $\mathcal{U}_{m+1}$ via cycle-consistency.

Let $N = |\mathcal{U}_{\backslash m}|$ and $\mathcal{U}_{\backslash m} = \{\boldsymbol{x}_i\}_{i=1}^N$, selecting $\mathcal{U}_{m+1} \subset \mathcal{U}_{\backslash m}$ is equivalent to selecting a binary indicator vector $\boldsymbol{q} \in \{0,1\}^N$, in which $q_i = 1^2$ means that $\boldsymbol{x}_i \in \mathcal{U}_{\backslash m}$ is included into $\mathcal{U}_{m+1}$. This representation turns the self-training step $\boldsymbol{\theta}_{m+1} = \texttt{ST}(\boldsymbol{\theta}_m,\mathcal{U}_{m+1})$ in Equation 8 into

$$\texttt{ST}(\boldsymbol{\theta}_m,\boldsymbol{q}) = \arg\min_{\boldsymbol{\theta}\in\Theta}\frac{1}{N}\sum_{i=1}^N q_i \times \ell\big(f(\boldsymbol{x}_i;\boldsymbol{\theta}),\texttt{sharpen}(f(\boldsymbol{x}_i;\boldsymbol{\theta}_m))\big), \tag{9}$$

We choose to solve Equation 8 approximately by relaxing the binary vector $\boldsymbol{q} \in \{0,1\}^N$ into a real vector $\boldsymbol{q} \in \mathbb{R}^N$, which fits perfectly into the meta-reweighting framework [29, 55]. Concretely, meta-reweighting treats $\boldsymbol{q} \in \mathbb{R}^N$ as learnable differentiable parameters associated to training examples (in our case, the data in $\mathcal{U}_{\backslash m}$). Meta-reweighting for Equation 8 can be implemented via the following six steps for multiple iterations.

1. Detach: $\boldsymbol{\theta} \leftarrow \boldsymbol{\theta}_m$,
2. Forward: $\boldsymbol{\theta}(\boldsymbol{q}) \leftarrow \boldsymbol{\theta} - \dfrac{\eta_{\boldsymbol{\theta}}}{|\mathcal{U}_{\backslash m}|} \times \dfrac{\partial \sum_{i\in\mathcal{U}_{\backslash m}} q_i \times \ell(f(\boldsymbol{x}_i;\boldsymbol{\theta}),\texttt{sharpen}(f(\boldsymbol{x}_i;\boldsymbol{\theta}_m)))}{\partial\boldsymbol{\theta}}$,
3. Detach: $\boldsymbol{\theta}' \leftarrow \boldsymbol{\theta}(\boldsymbol{q})$,
4. Backward: $\boldsymbol{\theta}(\boldsymbol{q}) \leftarrow \boldsymbol{\theta}(\boldsymbol{q}) - \dfrac{\eta_{\boldsymbol{\theta}}}{|\mathcal{U}_m|} \times \dfrac{\partial \sum_{j\in\mathcal{U}_m} \ell(f(\boldsymbol{x}_j;\boldsymbol{\theta}(\boldsymbol{q})),\texttt{sharpen}(f(\boldsymbol{x}_j;\boldsymbol{\theta}')))}{\partial\boldsymbol{\theta}(\boldsymbol{q})}$,
5. Update: $\boldsymbol{q} \leftarrow \boldsymbol{q} - \dfrac{\eta_{\boldsymbol{q}}}{|\mathcal{U}_m|} \times \dfrac{\partial \sum_{j\in\mathcal{U}_m} \ell(f(\boldsymbol{x}_j;\boldsymbol{\theta}(\boldsymbol{q})),\texttt{sharpen}(f(\boldsymbol{x}_j;\boldsymbol{\theta}_m)))}{\partial\boldsymbol{q}}$,
6. Update: $q_i \leftarrow \max\{0,q_i\}$.

---

[2]Here we use the same notation as the coarse scores defined in subsection 3.3, since later we will initialize these values indeed by the coarse scores.

The aforementioned updating rule gives $x_i$ a higher value of $q_i$ if it helps preserve the discriminative knowledge. After obtaining the updated $q \in \mathbb{R}^N$, we then sort it in the descending order and select the top $\frac{|\mathcal{U}|}{M-1}$ examples to be $\mathcal{U}_{m+1}$ (i.e., every intermediate domain has an equal size).

**Discussion.** The way we relax the binary-valued vector $q \in \{0, 1\}^N$ by real values is related to the linear programming relaxation of an integer programming problem. It has been widely applied, for example, in discovering sub-domains within a dataset [16]. Theoretically, we should constrain each element of $q$ to be within $[0, 1]$. Empirically, we found that even without the clipping operation to upper-bound $q_i$ (i.e., just performing $\max\{0, q_i\}$), the values in $q_i$ do not explode and the algorithm is quite stable: we see a negligible difference of using upper-bounded $q_i$ or not in the resulting accuracy of GDA. This is also consistent with the common practice of using meta-reweighting [29, 55].

**Initialization.** We initialize $q_i$ by the coarse scores defined in subsection 3.3. This is for one important reason: Equation 8 searches for the next intermediate domain $\mathcal{U}_{m+1}$ only based on the current intermediate domain $\mathcal{U}_m$. Thus, it is possible that $\mathcal{U}_{m+1}$ will include some data points that help preserve the knowledge in $\mathcal{U}_m$ but are distributionally deviated from the gradual transition between $\mathcal{U}_m$ and the target domain $\mathcal{T}$. The scores defined in subsection 3.3 thus serve as the guidance; we can also view Equation 8 as a refinement step of the coarse scores. As will be seen in section 4, the qualities of the initialization and the refinement steps both play important roles in IDOL's success.

# 4 Experiment

## 4.1 Setup

We mainly study two benchmark datasets used in [35]. Rotated MNIST [36] is a popular synthetic dataset to study distributional shifts. It gradually rotates the original $50,000$ images in the MNIST [37] dataset with $[0, 60]$ degrees uniformly, where $[0, 5)/[5, 55)/[55, 60]$ degrees are for the source/intermediate/target domains, respectively. However, since original images have no ground truths of the rotation, the rotation annotations are not expected to be perfect. Portraits dataset [14] is a real-world gender classification dataset of a collection of American high school seniors from the year 1905 to 2013. Naturally, the portrait styles change along years (e.g., lip curvature [14], fashion styles). The images are first sorted by the years and split. For both datasets, each domain contains $2000$ images, and $1,000$ images are reserved for both the source and target domains for validation.

We follow the setup in [35]: each model is a convolutional neural network trained for 20 epochs for each domain consequently (including training on the source data), using Adam optimizer [32] with a learning rate 0.001, batch size 32, and weight decay 0.02. We use this optimizer as the default if not specified. Hyper-parameters of IDOL include $K = 2M$ rounds for progressive training and 30 epochs of refinement per step (with mini-batch 128), where $M = 19$ for the Rotated MNIST and $M = 7$ for the Portraits. More details are in the supplementary material.

To study how the intermediate domain sequences affect gradual self-training, we consider baselines including "Source only" and "UDA" that directly self-trains on the target data ($\mathcal{T}$) and all the intermediate data ($\mathcal{U}$). We compare different domain sequences including *Pre-defined* indexes, e.g., *rotations* and *years*. We further consider the challenging setup that all intermediate data are unindexed. *Random* sequences are by dividing the unlabeled data into $M-1$ domains randomly, with self-training on the target in the end. For our IDOL, different domain scores introduced in subsection 3.3 are adopted for the coarse domain sequences, and we optionally apply cycle-consistency refinement to make them fine-grained.

## 4.2 Main study: comparison with pre-defined domain sequences

The main results on Rotated MNIST and Portraits are provided in Table 1. We first verify that without any domain sequences, UDA (on target and intermediate data) and GDA (on random sequences) do not perform well, though they slightly improve over the source model. On the contrary, training with pre-defined indexes that arguably guide the gradual adaptation significantly performs better.

**IDOL is competitive to pre-defined sequences.** IDOL can produce meaningful domain sequences (see Figure 3), *given only the intermediate unlabeled data without any order*. The domain discriminator with progressive training performs the best and is comparable to pre-defined sequences. The confidence scores by the classifier are not enough to capture domain shifts.

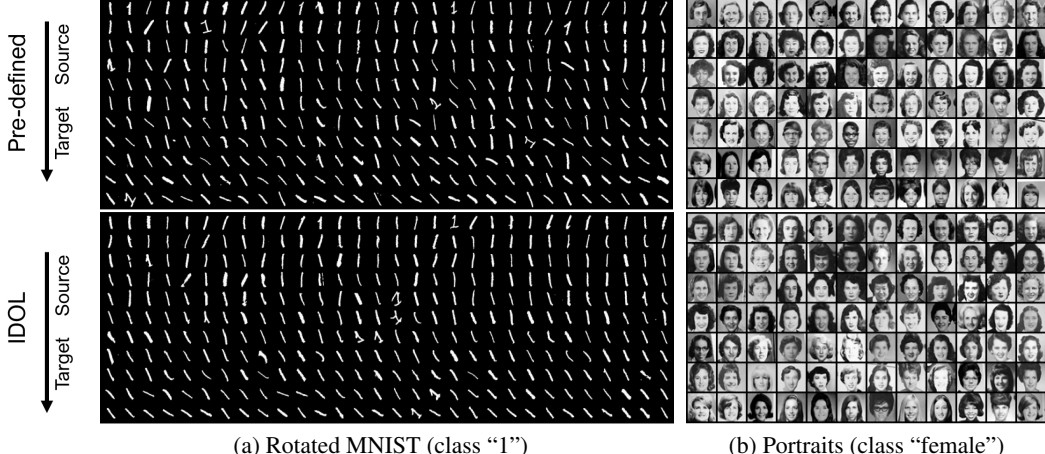

(a) Rotated MNIST (class "1")     (b) Portraits (class "female")

Figure 3: Random samples (the most representative class) from pre-defined indexes or IDOL sequences.

Table 1: Gradual self-training on Rotated MNIST and Portraits. Bottom section: IDOL with domain scores.

| Coarse scores | Indexed? | Adaptation | Refined? | Rotated MNIST | Portraits |
|---|---|---|---|---|---|
| None | ✗ | Source only | - | 31.9±1.7 | 75.3±1.6 |
| | | UDA ($\mathcal{T}$) | - | 33.0±2.2 | 76.9±2.1 |
| | | UDA ($\mathcal{T}+\mathcal{U}$) | - | 38.0±1.6 | 78.9±3.0 |
| Pre-defined [35] | ✓ | GDA | ✗ | 87.9±1.2 | 83.8±0.8 |
| | | | ✓ | 93.3±2.3 | 85.8±0.4 |
| Random | ✗ | GDA | ✗ | 39.5±2.0 | 81.1±1.8 |
| | | | ✓ | 57.5±2.7 | 82.5±2.2 |
| Classifier confidence | ✗ | GDA | ✗ | 45.5±3.5 | 79.3±1.7 |
| Manifold distance | | | ✗ | 72.4±3.1 | 81.9±0.8 |
| Domain discriminator | | | ✗ | 82.1±2.7 | 82.3±0.9 |
| Progressive domain discriminator | | | ✗ | 85.7±2.7 | 83.4±0.8 |
| | | | ✓ | 87.5±2.0 | 85.5±1.0 |

**Refinement helps and coarse initialization matters.** We observe that it is crucial to have high-quality coarse sequences as the initialization; the fine indexes by cycle-consistency refinement could further improve the coarse sequences, validating its effectiveness. We note that the Rotated MNIST dataset treats the original MNIST data as 0-degree rotation and artificially rotates the data given a rotation index. However, for data in the original MNIST dataset, there already exist variations in terms of rotations. This can be seen in Figure 3: in the first row of 0-degree rotation based on the pre-defined indexes, the examples of digit 1 do have slight rotations. Interestingly, in this situation, IDOL could potentially capture the true rotation of each example to further improve GDA.

**Analysis.** To understand why refinement helps, we take a closer look at the Portraits dataset that is naturally sorted by *years*. Are *years* the best factor to construct the domain sequence? We compare it with the refined sequence by IDOL. In Figure 4, we monitor the target accuracy of each step of adaptation and find that the accuracy fluctuates by using years. Interestingly, learning with IDOL stably and gradually improves the target accuracy and ends up with a higher accuracy, validating that

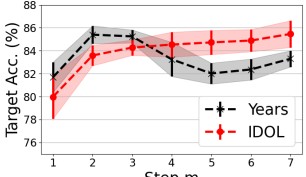

(a) Target Acc. of gradual ST.

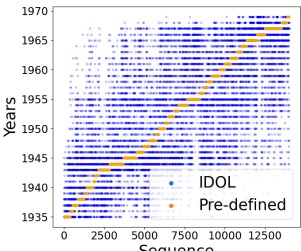

(b) Years vs. IDOL sequence.

Figure 4: Portraits with year indexes or IDOL domain sequence.

refinement with cycle-consistency indeed produces better sequences. Specifically, the IDOL sequence has a reasonable 0.727 correlation with *years* but they are not perfectly aligned. Several factors such as fashions, hairstyles, eyeglasses, and lip curvatures may not perfectly align with years [14]. Even within the same year, individuals could have variations in these factors as well. We thus hypothesize that IDOL can discover the domain sequence that reflects a smoother transition of these factors.

We note that the intermediate domains constructed by IDOL (domain scores with or without refinement) may not match the target "class" distributions. We monitor the number of examples of class $c$ in each domain $\mathcal{U}_m$, i.e., $|\mathcal{U}_{m,c}|$, and compute $\frac{1}{M-1}\sum_{m=1}^{M-1}\frac{\max_c\{|\mathcal{U}_{m,c}|\}}{\min_c\{|\mathcal{U}_{m,c}|\}}$, which ideally should be 1.0, i.e., class-balanced. We observe fine-grained indexes are indeed more class-balanced (coarse vs fine-grained): 1.33 vs 1.23 on Rotated MNIST and 1.27 vs 1.25 on Portraits.

### 4.3 Case study: learning with partial or outlier information

If partial or outlier information of the intermediate domains is given, is IDOL still helpful? We examine the question by three experiments. First, we consider *unsorted domains*: the data are grouped into $M-1$ domains according to the pre-defined indexes but the domains are *unordered*. We find that if we sort the domains with the mean domain scores $\frac{1}{|\mathcal{U}_m|}\sum_{\boldsymbol{x}_i\in\mathcal{U}_m}q_i$, then we can perfectly recover the pre-defined order. Second, we investigate if we have *fewer, coarsely separated pre-defined domains*: we double the size of $|\mathcal{U}_m|$. We find that on Rotated MNIST (w/ pre-defined indexes), the accuracy degrades by $11\%$. IDOL can recover the accuracy since IDOL outputs a sorted sequence of data points, from which we can construct more, fine-grained intermediate domains. Third, we investigate the addition of *outlier domains*, in which the starting/end intermediate domains do not match the source/target domains exactly. For example, on Rotated MNIST, the intermediate domains are expanded to cover $[-30, 90]$ degrees. GDA on such domain sequence with outlier domains degrades to $77.0 \pm 2.0$ accuracy. Refinement can still improve it to $81.3 \pm 1.4$.

### 4.4 Case study: learning with low-quality intermediate data or indexes

In practice, the intermediate unlabeled data or the pre-defined domain indexes for them might not be perfect. We consider several realistic cases to show that our method can make GDA more robust to the quality of the intermediate data or the domain sequences. First, we investigate *fewer intermediate data*: Equation 4 implies that the target error increases if the intermediate data are too sparse. We repeat the experiments in subsection 4.2 but with only $30\%$ intermediate data. Second, we consider *noisy indexes*: building upon $30\%$ clean indexes, we further add more data with noisy, random indexes.

Figure 5 summarizes the results. We first notice that with only $30\%$ intermediate data, even if the indexes are clean, the accuracy decreases drastically by $14\%$ for MNIST and by $5\%$ for Portraits. For MNIST, adding more noisily-indexed data actually decreases the accuracy. Portraits gains some improvements with the noisily-indexed data, suggesting that the task could benefit from learning with intermediate data (even without accurate indexes). One advantage of IDOL is that we can annotate domain indexes for any unindexed data. On both datasets, IDOL stably improves with more data available, significantly outperforming that without IDOL.

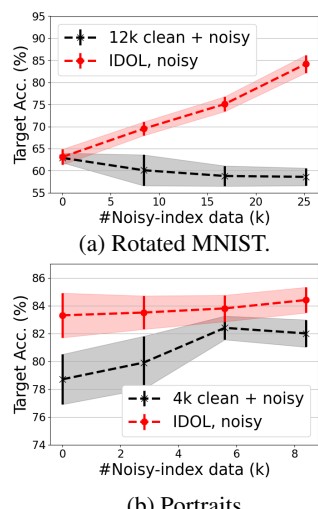

(a) Rotated MNIST.

(b) Portraits.

Figure 5: Different numbers of noisily-indexed intermediate data.

### 4.5 Case study: CIFAR10-STL UDA with additional, open-domain unlabeled data

We examine IDOL on a CIFAR10-to-STL [5, 34] UDA task which is known to be challenging due to the small data size [4]. The STL dataset has an additional unlabeled set, whose data are sub-sampled from ImageNet [9]. We use this set as the intermediate data, which contains unknown or outlier domains and classes. Here, we do not aim to compete with the state of the arts on the CIFAR10-to-STL UDA task, but study a more general application scenario for IDOL.

Table 2: CIFAR10-STL.

| Method | Target Acc. |
|---|---|
| Source only | $76.6\pm0.4$ |
| UDA ($\mathcal{T}$) (lr $=10^{-4}$) | $69.4\pm0.4$ |
| UDA ($\mathcal{T}$) (lr $=10^{-5}$) | $75.1\pm0.3$ |
| UDA ($\mathcal{T}+\mathcal{U}$) | $61.1\pm0.8$ |
| GDA w/ conf. | $77.1\pm0.5$ |
| GDA w/ IDOL | $78.1\pm0.4$ |

To leverage these data, we use $M=3$ and filter out $80\%$ of them using the classifier confidence. We first train a ResNet-20 [23] source model. In Table 2, we observe that direct UDA using self-training on STL or unlabeled data are inferior to the source model, even if the learning rate is carefully tuned. GDA with IDOL achieves the highest accuracy, demonstrating its robustness and wide applicability.

# 5 Related Work

**Domain adaption.** UDA has been studied extensively [1, 53, 63], and many different approaches have been proposed such as minimizing domain divergence [13, 20, 45, 61, 62, 65], cross-domain [39, 57, 58, 61] and domain-invariant features [18, 26, 54, 67–69, 73]. Self-training recently emerges as a simple yet effective approach [2, 28, 30, 31, 41, 64, 77]. It uses the source model to provide pseudo-labels for unlabeled target data, and approaches UDA via supervised learning in the target domain [38, 48, 49]. Theoretical analyses for applying self-training in UDA are provided in [3, 70].

**Gradual domain adaption.** Many UDA works show the benefits of gradually bridging the domain gap. With features as input, several work [8, 15, 19] construct Grassmannian manifolds between the source and the target. For deep models, progressive DA [27] proposes to create the synthetic intermediate domain with an image-to-image translation network. Other works [7, 17] instead generate intermediate data with jointly-trained generators. Na et al. [51] augments the intermediate data with Mixup [74]. Unlike all the above work that focus on building synthetic intermediate domains given only the source and target data, gradual domain adaption proposed in [12, 25, 35, 71] studies how to leverage extra real intermediate data with pre-defined domain indexes.

**Learning with cycle consistency.** The concept of cycle consistency has been successfully applied in many machine learning tasks. On a high level, cycle supervision enforces that the translation to the target should be able to be translated back to match the source. For instance, back-translation [22, 59] is a popular technique to perform unsupervised neural translation in natural language processing. Many computer vision tasks such as image generation [56], image segmentation[40], style transfer [76], etc., can also be improved by matching the structural outputs with cyclic signals. We propose a novel approach that uses cycle consistency to discover intermediate domains for gradual adaption.

**Discovering feature subspaces.** We further discuss the connections of our work to existing efforts about subspace learning. One powerful technique is subspace clustering [10, 33, 52] that partitions data into many subspaces (e.g., classes) unsupervisedly. However, it may not be applied to GDA directly since it groups with discriminative features which might not capture domain shifts. Specifically for UDA, some works aim to discover sub-spaces in the source/target domains [16, 24, 43, 47, 72]. The assumption is that the datasets might not be collected from one environment but generated from many different distributions, re-formulating it as a multi-domain adaption problem. We note that, the sub-domains discovered in these methods are fundamentally different from the domain sequence we discover. The sub-domains are assumed to have a dedicated adaption mapping to the target domain. The domain sequences in GDA are for constructing a path from the source to the target via intermediate data. How to link these learning techniques to GDA will be interesting future work.

# 6 Conclusion

Gradual domain adaptation leverages intermediate data to bridge the domain gap between the source and target domains. However, it relies on a strong premise — prior knowledge of domain indexes to sort the intermediate data into a domain sequence. We propose the IDOL algorithm that can produce comparable or even better domain sequences without pre-defined indexes. With a simple progressively-trained domain discriminator, it matches the performance of using pre-defined indexes, and further improves with the proposed cycle-consistency refinement. Essentially, IDOL is a useful tool to augment any GDA algorithms with high-quality domain sequences given unindexed data.

## Acknowledgments and funding transparency statement

This research is partially supported by NSF IIS-2107077 and the OSU GI Development funds. We are thankful for the generous support of the computational resources by the Ohio Supercomputer Center.

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
