# Supplementary Material

We provide details omitted in the main paper.

- section 7: more details of IDOL (cf. section 3 of the main paper).
- section 8: details of implementations and experimental setups, computation cost, more results for refinement in IDOL (cf. section 4 of the main paper).
- section 9: broader impact and potential negative societal impacts of our work.
- section 10: limitations and future work.

## 7  Details of IDOL

### 7.1  Another view for the coarse-to-fine framework of IDOL

Another way to understand the coarse-to-fine framework we proposed is from the optimization perspective. In Equation 3 of the main paper, we provide the overall objective of IDOL, which is however intractable due to (a) no labeled target data to evaluate the domain sequence and (b) the combinatory nature of searching for a domain sequence. We deal with (a) by Equation 7 of the main paper. That is, we propose the cycle-consistency loss to measure the quality of the domain sequence, which requires no labeled target data. Since Equation 7 is still hard to solve, we relax it by a greedy approach. We find one domain at a time in sequence, starting from $\mathcal{U}_1$ (i.e., the one closest to the source) to $\mathcal{U}_{M-1}$ (i.e., the one closest to the target). Each sub-problem is described in Equation 8 of the main paper. The relaxation may lead to sub-optimal solutions. Concretely, each sub-problem is not aware of other already selected intermediate domains or other future intermediate domains to be selected. To mitigate this issue, we propose to assign each data point a coarse domain score (i.e., subsection 3.3 of the main paper), which serves as the initialization of each sub-problem.

### 7.2  Algorithms

Here we provide the summary of the IDOL algorithm. IDOL learns to sort the unindexed intermediate data to a sequence, partitioned into several intermediate domains. An illustration is provided in Figure 6. As shown in algorithm 1, there are three main steps in the overall procedure: first, we construct the coarse domain sequence by learning to predict the domain score for each example and sorting the examples according to the domain scores. Second, we refine the coarse indexes with cycle-consistency as shown in algorithm 2. The refinement is decomposed into several steps, gradually discovering the *next* intermediate domain in sequence. Each step is to refine the coarse indexes with meta-reweighting [29, 55] and takes the closest examples to the current domain as the next domain, as shown in algorithm 3. Finally, IDOL outputs a sequence of intermediate data points; it can then be divided into several intermediate domains for gradual domain adaption.

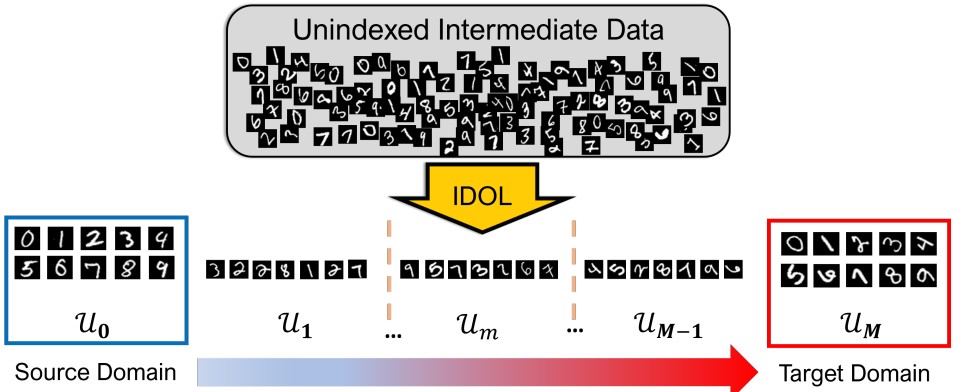

Figure 6: **Gradual domain adaption (GDA) without indexed intermediate domains.** Our IDOL algorithm sorts the unindexed intermediate data into a sequence, from the source domain to the target domain, then it can be further partitioned into several intermediate domains for gradual domain adaption.

**Algorithm 1:** Intermediate **DO**main **L**abeler (**IDOL**)

**Input:** Labeled source data $\mathcal{S}$, unlabeled target data $\mathcal{T}$, intermediate data $\mathcal{U}$, and # of domains $M-1$;

1  **Coarse indexing (by progressive training for the domain discriminator):** learn $g(\cdot;\boldsymbol{\phi})$ with $\mathcal{S},\mathcal{T},\mathcal{U}$ and assign a score $q_i = g(\boldsymbol{x}_i^{\mathcal{U}};\boldsymbol{\phi})$ to every $\boldsymbol{x}_i^{\mathcal{U}} \in \mathcal{U}$ (cf. subsection 3.3 in the main paper);

2  **Construct:** indexed sequence $I_{\text{coarse}} = (\boldsymbol{x}_1,...,\boldsymbol{x}_{|\mathcal{U}|})$ by sorting $\{q_i = g(\boldsymbol{x}_i^{\mathcal{U}};\boldsymbol{\phi})|\forall\boldsymbol{x}_i^{\mathcal{U}} \in \mathcal{U}\}$;

3  **Fine-grained indexes:** learn $I_{\text{fine-grained}}$ with refinement (algorithm 2);

4  **Construct:** domain sequence by chunking $I_{\text{fine-grained}}$ into $M-1$ domains;

  **Output:** $(\mathcal{U}_1,...,\mathcal{U}_{M-1})$.

---

**Algorithm 2:** Refinement of the coarse sequence

**Input:** Labeled source data $\mathcal{S}$, # of domains $M-1$, index sequence $I_{\text{coarse}} = (\boldsymbol{x}_1,\ldots,\boldsymbol{x}_{|I_{\text{coarse}}|})$.

**Initialize:** Pre-train the source model $\boldsymbol{\theta}_0$ on $\mathcal{S}$, $\mathcal{S}_0 \leftarrow \mathcal{S}$, chunk size $C = \frac{|I_{\text{coarse}}|}{M-1}$;

1  **for** $m \in [0,1,\ldots,M-2]$ **do**

2     **Initialize:** data parameter $\boldsymbol{q} = [\frac{|I_{\text{coarse}}|}{|I_{\text{coarse}}|}, \frac{|I_{\text{coarse}}|-1}{|I_{\text{coarse}}|}, \ldots, \frac{0}{|I_{\text{coarse}}|}]$;

    Obtain the next domain with $I_{m+1} \leftarrow \texttt{FindNextDomain}(\boldsymbol{\theta}_m, I_{\text{coarse}}, \mathcal{S}_m, \boldsymbol{q}, C)$;       //algorithm 3

3     Pseudo-label $I_{m+1}$ to construct $\mathcal{S}_{m+1} = \{(\boldsymbol{x}_i, \texttt{sharpen}(f(\boldsymbol{x}_i, \boldsymbol{\theta}_m)))\}_{\boldsymbol{x}_i \in I_{m+1}}$;

4     $\boldsymbol{\theta}_{m+1} \leftarrow$ self-train $\boldsymbol{\theta}_m$ on $\mathcal{S}_{m+1}$;

5     Update $I_{\text{coarse}} \leftarrow (\boldsymbol{x}_i|\boldsymbol{x}_i \in I_{\text{coarse}}, \boldsymbol{x}_i \notin I_{m+1})$;

  **Output:** Concatenate $I_1,\ldots,I_{M-1}$ as the fine indexes $I_{\text{fine-grained}}$.

---

**Algorithm 3:** Finding the next domain with cycle-consistency (FindNextDomain)

**Input:** $\boldsymbol{\theta}_m$, (pseudo-)labeled data $\mathcal{S}_m = \{(\boldsymbol{x}_i, y_i)\}_{i=1}^{|\mathcal{S}_m|}$, intermediate index sequence $I = (\boldsymbol{x}_1,...,\boldsymbol{x}_{|I|})$, initial data parameters $\boldsymbol{q} \in \mathbb{R}^{|I|}$ ($q_i$ is corresponded to $\boldsymbol{x}_i$ in $I$), loss function $\ell$, learning rate $\eta_{\boldsymbol{\theta}}, \eta_{\boldsymbol{q}}$, and chunk size $C$.

1  **while** *stop* **do**

2     Detach $\boldsymbol{\theta}^{(0)} \leftarrow \boldsymbol{\theta}_m$;

3     **for** $t \in [1,\ldots,T]$ **do**

4         Sample a mini-batch $B_{\mathcal{U}}$ from $I, \boldsymbol{q}$;               //Forward adaption.

5         $\boldsymbol{\theta}^{(t)} \leftarrow \boldsymbol{\theta}^{(t-1)} - \eta_{\boldsymbol{\theta}} \nabla_{\boldsymbol{\theta}^{(t-1)}} \sum_{i \in B_{\mathcal{U}}} q_i \times \ell(f(\boldsymbol{x}_i; \boldsymbol{\theta}^{(t-1)}), \texttt{sharpen}(f(\boldsymbol{x}_i; \boldsymbol{\theta}_m)))$;

6     Detach $\boldsymbol{\theta}' \leftarrow \boldsymbol{\theta}^{(T)}(\boldsymbol{q})$;

7     **for** $t \in [T,\ldots,2T-1]$ **do**

8         Sample a mini-batch $B_{\mathcal{S}_m}$ from $\mathcal{S}$;              //Backward adaption.

9         $\boldsymbol{\theta}^{(t+1)} \leftarrow \boldsymbol{\theta}^{(t)} - \eta_{\boldsymbol{\theta}} \nabla_{\boldsymbol{\theta}^{(t)}} \frac{1}{|B_{\mathcal{S}_m}|} \sum_{i \in B_{\mathcal{S}_m}} \ell(f(\boldsymbol{x}_i; \boldsymbol{\theta}^{(t)}), \texttt{sharpen}(f(\boldsymbol{x}_i; \boldsymbol{\theta}')))$;

10    Sample a mini-batch $B_{\mathcal{S}_m}$ from $\mathcal{S}_m$;     //Update data parameters with cycle-consistency.

11    Update $\boldsymbol{q} \leftarrow \boldsymbol{q} - \eta_{\boldsymbol{q}} \nabla_{\boldsymbol{q}} \frac{1}{|B_{\mathcal{S}_m}|} \sum_{i \in B_{\mathcal{S}_m}} \ell(f(\boldsymbol{x}_i; \boldsymbol{\theta}^{(2T)}(\boldsymbol{q})), y_i)$;

12    $q_i \leftarrow \max\{0, q_i\}, \forall q_i \in \boldsymbol{q}$;

13  $I \leftarrow$ sort $I$ by $\boldsymbol{q}$ (descending order);

  **Output:** the next domain $I[:C]$.

# 8  Implementation Details

## 8.1  Experimental setup

**Dataset, model, and optimizer.** The Rotated MNIST and Portraits datasets are resized to $28 \times 28$ and $32 \times 32$, respectively, without data augmentation. In the CIAFR10-STL experiments, images are resized to $32 \times 32$. We use the standard normalization and data augmentation with random horizontal flipping and random cropping as in [23].

For the Rotated MNIST and Portraits datasets, we adopt the same network used in [35]. The network consists of 3 convolutional layers, each with the kernel size of 5, stride 2, and 32 channel size. We use ReLU activations for all hidden layers. After the convolutional layers, it follows with a dropout layer with 0.5 dropping rate, a batch-norm layer, and the Softmax classifier. We use the Adam optimizer [32] with the learning rate 0.001, batch size 32, and weight decay 0.02. We use this

Table 3: IDOL with refinement on different coarse domain scores.

| Coarse scores | Indexed? | Adaptation | Refined? | Rotated MNIST | Portraits |
|---|---|---|---|---|---|
| Classifier confidence | | | ✗ | 45.5±3.5 | 79.3±1.7 |
| | | | ✓ | 62.5±2.1 | 83.6±1.6 |
| Manifold distance | ✗ | GDA | ✗ | 72.4±3.1 | 81.9±0.8 |
| | | | ✓ | 82.4±2.3 | 85.2±0.9 |
| Domain discriminator | | | ✗ | 82.1±2.7 | 82.3±0.9 |
| | | | ✓ | 86.2±2.2 | 85.1±1.3 |
| Progressive domain discriminator | | | ✗ | 85.7±2.7 | 82.3±0.9 |
| | | | ✓ | 87.5±2.0 | 85.5±1.0 |

optimizer and train for 20 epochs for the Rotated MNIST and Portraits datasets as the default if not specified, including training the source model, self-training adaption on each domain, our domain discriminator, and progressive training for each step.

For the CIFAR10-STL experiments, we train a ResNet-20 for 200 epochs for the source model and 80 epochs for both the domain discriminator and progressive training for each step with Adam optimizer with learning rate 0.00001, batch size 128, and weight decay 0.0001.

We use the same network for our domain discriminator but replace the classifier with a Sigmoid binary classifier.

**GDA.** For GDA, we focus on gradual self-training studied in [35]. We follow the common practice to filter out low confidence pseudo-labeled data for self-training on every domain. That is, in applying Equation 1 of the main paper for self-training, we only use data with high prediction confidences. We keep top $90\%$ confident examples for the Rotated MNIST and Portraits datasets and top $20\%$ confident examples for the CIFAR10-STL experiments due to the fact that most of the unlabeled data are noisy and could be out-of-distribution. Each domain is trained for 20 epochs for the Rotated MNIST and Portraits datasets. We train 80 epochs for the CIFAR10-STL experiments.

**IDOL.** For hyperparameters specific to IDOL, we tune with the target validation set. We did not change hyperparameters in GDA for fair comparison. We set $K = 2M$ rounds for progressive training. For algorithm 3, we set $T = 10$ and train for in total 30 epochs for all datasets, with batch size 128. The learning rates are set as $\eta_{\boldsymbol{\theta}} = \eta_{\boldsymbol{q}} = 0.001$.

## 8.2 Computation cost

We run our experiments on one GeForce RTX 2080 Ti GPU with Intel i9-9960X CPUs. In algorithm 3, updating the data parameter $\boldsymbol{q}$ requires it to backward on gradients of $\boldsymbol{\theta}$, which approximately takes three-time of the computation time of a standard forward-backward pass [55].

We estimated the time consumption on the Portraits dataset experiments. For coarse scores, the confidence is simply by applying the classifier to the input data. Calculating the manifold distance via [50] takes about 20 seconds with Intel i9-9960X CPU. The domain discriminator (both w/ or w/o progressive training) and the fine stage involve GPU computation. Using a GeForce RTX 2080 Ti GPU, training a domain discriminator takes about 4 seconds. Progressive training takes about $M$ times longer, where $M$ is the number of intermediate domains (which is 7 here). This is because it trains the domain discriminator for $M$ rounds. For the refinement stage proposed in cf. subsection 3.4 based on meta-learning, Discovering one intermediate domain takes about 44 seconds and we perform it for $M$ rounds.

## 8.3 More results on refinement

We provide the full experiments (cf. Table 1 in the main paper) of IDOL refinement on different coarse domain scores in Table 3. We observe the refinement consistently helps on the coarse domain scores, and the quality of the coarse domain scores is important to the final performance.

Next, we design the following experiment to investigate the variations of different trials and the effects of the number of intermediate samples. We apply IDOL (with progressive discriminator and refinement) for 10 runs with different random seeds, and record the intermediate domain index (from source $= 0$ to the target $= M$) to which each sample is assigned. For instance, if a sample is assigned to the second intermediate domain, we record 2 for this sample. After 10 runs, each sample will end up with 10 indices, and we calculate the variance. If a sample is always assigned to the same intermediate domain, the variance will be zero for this sample. We repeat the experiments with

Table 4: Variances of domain assignments.

| Intermediate data | Rotated MNIST | Portraits |
|---|---|---|
| 100% | 1.12 | 0.147 |
| 50% | 1.33 | 0.150 |
| 25% | 1.35 | 0.136 |

100%/50%/25% of intermediate data (dropped randomly). The averaged variance over all samples for both the Rotated MNIST dataset ($M = 19$) and Portraits dataset ($M = 7$). As shown in Table 4, the domain sequences formed in different trials are pretty consistent since the variances are small. Rotated MNIST has a higher variance than Portraits probably because the number of intermediate domains is larger. Besides, the amount of the intermediate data does not significantly affect the consistency among trials.

## 9   Broader Impact

Unsupervised domain adaption aims to address the distribution shift from the labeled training data to the unlabeled target data. Our work focuses on leveraging extra unlabeled data to help unsupervised domain adaption. We only assume that the unlabeled data are generated from underlying distributions that can be viewed as structured shifts from the source to the target domain. To the best of our knowledge, our algorithm would not cause negative effects on data privacy, fairness, security, or other societal impacts.

## 10   Limitations and Future Work

Nearly all the domain adaptation algorithms need to make assumptions on the underlying data distributions. For our algorithm IDOL, we generally require the additional unlabeled data to distribute in between the source and the target domains, which is the standard setup of gradual domain adaptation (cf. section 2 of the main paper). GDA will be less effective if the additional unlabeled data besides the source and target domains do not gradually bridge the two domains or contain outliers. If there are unlabeled data close to the source or target domains but are not distributed in between of them (e.g., if the source/target are digits with 60/0-degree rotation, then digits with 70-degree are one of such cases), IDOL may still include those data into the intermediate domains and could potentially degrade the overall performance. In subsection 4.5 of the main paper, we further show that, even if the additional unlabeled data contain outliers or does not smoothly bridge the two domains, IDOL can still effectively leverage the data to improve the performance on the target domain.

Our current scope of the experiments is mainly limited by the availability of benchmark datasets for gradual domain adaptation. This is mainly because GDA is a relatively new setting. Most of the existing datasets for domain adaptation are designed for the conventional unsupervised domain adaptation, in which only the source and target domain data are provided. Given (1) the superior performance of gradual domain adaptation by leveraging the additional unlabeled data and (2) the fact that real-world data change gradually more often than abruptly, we believe it will be useful to develop more datasets for gradual domain adaptation. For instance, datasets in autonomous driving that involve shifts in time and geo-locations could be an option. Other examples include sensor measurements drift over time, evolving road conditions in self-driving cars, and neural signals received by brain-machine interfaces, as pointed out in the introduction of [35].