# OpenReview forum: "Gradual Domain Adaptation without Indexed Intermediate Domains"
_NeurIPS.cc/2021/Conference — NeurIPS 2021 Poster_

### Official Review · Reviewer_B4p7 · 2021-07-07

**Rating:** 7
**Confidence:** 4

**Summary:**

This paper is addressing the gradual domain adaptation (GDA) problem without the access of intermediate domain indices. This paper proposes a coarse-to-fine method to predict the indices of the intermediate domain sequence. The results are shown to be comparative to pre-defined methods.

**Limitations And Societal Impact:**

Yes, but it would be better to discuss more limitations as mentioned above.

**Main Review:**

1. claim concerns:
a) One of the main concerns is the definition of “intermediate domain” in this paper. It highly depends on how the intermediate domains are generated. This paper only briefly mentions that the assumption is that the intermediate domains are under the data distribution between source and target domains. It would be great to have more analysis about: 1) What if the distributions of intermediate domains are not gradually/smoothly underlying between source and target domains, but closer to on side instead? 2) What if the intermediate domains are under very different distributions? With the above analyses, it will be easier to clarify in what situation the proposed method works.

2. technical detail concerns:
a) The descriptions of “progress training for the domain discriminator” are not clear (lines 169-180). It would be clearer if there are pseudo codes or algorithms like the one in Sec. 3.4 (line 220-225).

3. experiment concerns:
a) In Table 1, does “Refined?” mean the fine stage as described in Sec. 3.4? If so, why does this refinement also work for the “pre-defined” case? It would be great to analyze the pre-defined case with and without refinement (i.e., visualized Table 1 using a similar way as Figure 4).
b) Sec. 4.5 partially answers the question about “What if the intermediate domains are under very different distributions?”. From Table 2, it shows that the proposed method does not work well for this case. It would be better to have more analysis and explanation here.
Moreover, the result of “UDA (T)” is weird. The numbers in Table 2 are not consistent with those in RCA [4], which shows a 81.6 accuracy as “UDA (T)” and also outperforms “Source only”.

4. other concerns:
a) The proposed method is like an additional step to generate predicted domain indices, so it would be better to have time analysis to show the computation cost. The paper only briefly mentions that using short descriptions. It would be better to have more details (e.g., a Table with numbers).


**Time Spent Reviewing:**

6

---

> ### Author Response · Authors · 2021-08-10
> **Thank you for your valuable comments! Response to Reviewer B4p7**
>
> We thank the reviewer for the valuable comments.
>
> **Claim concerns.** For gradual domain adaptation (GDA), we follow the definition of intermediate domains given in Section 3 of [30]. Our Section 2.2 provides a summary of the definition. Concretely, for GDA to be effective, every two consecutive intermediate domains should have distributions close to each other, e.g., measured by the Wasserstein-infinity distance. In this paper, we assume that the intermediate domain sequence is not readily available and we aim to discover it. More specifically, we aim to discover the sequence to maximize the performance of the GDA algorithm (cf. Equation 5).
>
> We thank the reviewer for the suggestion on the analysis. Theoretically, according to Equation 4 in our paper and Section 3 in [30], (1) if the intermediate domains are not smoothly underlying between source and target domains or (2) if the intermediate domains are under very different distributions from the source and target domains, $\rho$ and $\beta$ in Equation 4 will be large, leading to poor performance for GDA.
>
> We have empirically verified (2) in the CIFAR10-STL experiment in Section 4.5. To empirically verify (1), we consider several experiments using GDA with the pre-defined indexes. We first chunk the intermediate data into domains, then consider several settings including:
>
> * Source/target-side: as suggested by the reviewer, we keep only those intermediate domains close to the source or the target.
> * Shuffle: we randomly shuffle the intermediate domains so that they are not smoothly changing.
>
> The target accuracies (%) are below:
>
> | Settings  | Rotated MNIST | Portraits   |
> | :---        |    :----:   |          ---: |
> | Pre-defined  | 87.9$\\pm$1.2       | 83.8$\\pm$0.8  |
> | Source-side  |   70.2$\\pm$1.8   | 82.4$\\pm$1.1 |
> | Target-side |    68.8$\\pm$2.0    |  81.9$\\pm$1.4 |
> | Shuffle    | 38.2$\\pm$2.1       | 81.6$\\pm$2.0  |
>
> Compared to the original sequences, the source/target-side settings drop as expected. The shuffle setting drops more significantly on both datasets.
>
>
> **Technical detail concerns.** Thank you for the suggestion. We provide the pseudo-code for progress training for the domain discriminator (Line 169-180) as follows. We will include a more detailed version in the final version.
>
> Given the source data $S$, the target data $T$, the intermediate data $U$, and the number of intermediate domains $2K$, we aim to progressively train a domain discriminator that can predict the domain score $q= g(x, \phi)$ for an input $x$. The larger the score is, the closer the example is to the source domain. Let $N$ denote the size of $U$.
>
> For $k$ from 1 to $K$:
> 1. Train $g(\cdot, \phi)$ using $S$ and $T$.
> 2. Predict $q_i = g(x_i, \phi), \forall x_i \in U$.
> 3. Rank all $q_i$ descendingly.
> 4. Those with the largest $\frac{N}{2K}$ scores form a new intermediate domain for the source side (index = $\frac{2K-k}{2K}$). We remove them from $U$ and add them into $S$.
> 5. Those with the smallest $\frac{N}{2K}$ scores form a new intermediate domain for the target side (index = $\frac{k}{2K}$). We remove them from $U$ and add them into $T$.
>
> **Experiment concerns.**
>
> **(a):** Yes, “Refined?” in Table 1 means the fine stage as described in Section 3.4. We will clarify this in the caption. The reason why the fine stage can further improve the pre-defined sequence is that the pre-defined sequence is not necessarily optimal for GDA (Line 47-49). For instance, in the Portraits dataset [12], the domain changes between the source and target domains result from several factors such as fashions, hairstyles, eyeglasses, and lip curvatures that may not perfectly align with years (please see [12] for discussions). Even within the same year, individuals could have variations in these factors as well. Thus, the pre-defined sequence by years may be sub-optimal, and IDOL can potentially discover the domain sequence that reflects a smoother transition of these factors than years. The Rotated MNIST dataset treats the original MNIST data as 0-degree rotation, and artificially rotates the data given a rotation index. However, for data in the original MNIST dataset, there already exist variations in terms of rotations. This can be seen in Figure 3: in the first row of 0-degree rotation based on the pre-defined index, the examples of digit 1 do have slight rotations. In this situation, IDOL could potentially capture the true rotation of each example. We will provide further analysis in the final version. We note that Figure 4 (b) did compare the pre-defined sequence and the refined sequence for the Portraits dataset. We will include a figure for the Rotated MNIST in the final version.
>
> **(b):** We conducted the CIFAR10-STL experiment mainly to investigate the challenges and limitations of both GDA and IDOL when the intermediate data are from a dissimilar distribution and do not smoothly bridge the source and target domains. As mentioned in Line 326, we do not aim to compete with the state-of-the-art UDA results. For UDA($T$), we are referring to the self-training method, which is the base algorithm for GDA in both [30] and our paper and we will clarify this. It is possible that some other UDA methods can lead to higher accuracy. Since the intermediate data are from a dissimilar distribution and do not smoothly bridge the source and target domains, self-training on all the data could not improve the source model (i.e., the UDA ($T + U$) row in Table 2). By using IDOL to discover the intermediate domains and filtering out some out-of-distribution data using the confidence, we can improve upon the source model, but the improvement is much smaller than that in Table 1.
>
> **Other concerns on the computation cost.** Thank you for your suggestion. We provide an estimation of the real-time cost of the coarse stage and fine stage in the **general response** above. We will organize them into a table in the final version.
>
>
> **Limitations and societal Impact.** Thanks for the suggestion. We will incorporate the responses above to strengthen the description of the limitation in the final version.

---

> > ### Comment · Reviewer_B4p7 · 2021-08-18
> > **Generalization ability concerns for the proposed method**
> >
> > The authors address all my concerns about technical details via the rebuttal. However, the proposed method can only deal with the GDA problem which has discoverable intermediate domains. The datasets used in this paper are relatively easy. Most standard DA datasets (e.g., Office, Office-Home, VisDA) are not evaluated. This raises doubt about the generalization ability of the proposed method to larger-scale datasets.

---

> > > ### Author Response · Authors · 2021-08-23
> > > **Re: Generalization ability concerns for the proposed method**
> > >
> > > We thank the reviewer for the positive feedback about our rebuttal. We are glad that we have addressed all your concerns about technical details. We also appreciate your additional comments.
> > >
> > > We want to reiterate that the current setup of gradual domain adaptation (GDA), in which the intermediate domains should be discovered and indexed properly with gradual domain shifts, is given by existing papers like [30]. The main purpose of our paper is to strengthen GDA's applicability, even when the intermediate domains are not yet separated and indexed. We agree that how to extend the scope of GDA, for example, to leverage unlabeled data that do not gradually shift between domains, is an interesting direction to explore, and we will leave it as our future work. That being said, we also believe in many real cases the data distribution does gradually shift, especially when they are collected over time or over consecutive locations. Examples include sensor measurements drift over time, evolving road conditions in self-driving cars, and neural signals received by brain-machine interfaces, as pointed out in the introduction of [30].
> > >
> > > About the experiments, our current study is mainly limited by the availability of benchmark datasets for gradual domain adaptation. This is mainly because GDA is a relatively new setting. Most of the existing datasets for domain adaptation (like Office, Office-Home, VisDA, as the reviewer mentioned) are designed for unsupervised domain adaptation (UDA), in which only the source and target domain data are provided. In other words, they are not directly applicable for GDA. We appreciate your concern. We note that even for UDA, its development began with a few small-scale datasets and gradually extended to more, large-scale datasets after attracting increasing attention. Given (1) the superior performance of gradual domain adaptation and (2) the fact that real-world data change gradually more often than abruptly, we believe that more and more datasets for gradual domain adaptation will be developed and released in the near future.

---

> > > > ### Comment · Reviewer_B4p7 · 2021-08-26
> > > > **Thank you for the detailed response**
> > > >
> > > > I appreciate the detailed response by the authors.
> > > > My main concern about the scope of experiments is overall addressed, so I would like to increase my rating to "accept".
> > > > However, I strongly recommend the authors add more explanations about the limitations of the current experiment setting (GDA) and discuss the possible ways to actively address these issues. For example, instead of waiting for more datasets to be released, it would be much better to propose a new dataset to increase the scope and scale of the GDA problem. I believe these kinds of discussions will greatly benefit the community for this research direction.

---

### Official Review · Reviewer_New7 · 2021-07-17

**Rating:** 7
**Confidence:** 4

**Summary:**

This paper is concerned with gradual domain adaptation, when we have access to data from intermediate distributions between source and target and use them to gradually adapt the model through self-training. To address the challenge of lack of intermediate data annotated and grouped based on their distance to the source domain, the paper proposes IDOL (Intermediate Domain Labeler) to index unlabeled available data that presumably covers the gap between source and target  domains based on their distance to source.

IDOL consists of two steps:
1. A progressively trained domain discriminator is used to assign a score to each example. (a higher/lower score indicates that the example is closer to the source/target domain).
2. A model based on a cycle-consistency loss is applied to refine the coarse scores progressively. The main idea is to group examples into intermediate domains such that enough discriminative information is preserved that we could approximately regain the accuracy on the source domain by consuming the intermediate domains in the reverse direction..

**Empirical results**:
Experiments on  Rotated MNIST and Portraits indicate that with IDOL we can achieve comparable performance on gradual domain adaptation as when we have the ground truth (pre-defined) domain sequences. Furthermore the experiments indicate that applying IDOL on pre-defined domain sequences, treating them as the outputs from step 1, and refining them through step 2 can lead to domain sequences that are better suited for gradual domain adaptation. In general, it seems that applying IDOL in many cases with partial or noisy annotation of the intermediated domain sequences can be beneficial.  Finally, IDOL is also compared with UDA on CIFAR10 to STL tasks, indicating superiority of both (GDA + confidence) and (GDA + IDOL) to UDA in this setting.


**Limitations And Societal Impact:**

I don't think the authors have adequately addressed the limitations of the proposed method (or about negative societal impact ).
I think there might be some room for discussion there. E.g., could this gradual process results in some unintentional biases in the models?



**Main Review:**

**Originality**: The proposed idea builds up on top of  existing works on gradual domain adaptation that study how leveraging samples from intermediate distributions can help in the unsupervised domain adaptation process. The idea is novel and theoretically motivated. It is a two step process of forming a sequence of domains given a set of unlabeled data points.

**Quality and Clarity**: I find the paper well written and very readable. The experiments and analysis provided in the paper validate the success of the IDOL in labeling intermediate sequence both qualitatively and quantitatively, but in a limited scope.

**Significance**: Based on the empirical results provided in the paper, the method seems to lead to significant improvements over the baselines it is compared with. However, both the baselines and the benchmarks are not selected generously. Even though the experiments are limited to a few small datasets, I can understand if this is mainly due to lack of benchmarks that would be applicable for the purpose of this study. There might be some other applications for the proposed approach to make sequences of domains  from a set of unlabeled examples. E.g., for analytical studies.

**Question for the authors:**
- Have you looked into the variations in the domain sequences formed in different trials? and maybe different factors that would affect them? e.g. number of intermediate samples? nature of the shift in distribution? (e.g., comparing rotated MNIST with the Face dataset).

**Main weakness(es)**:
- Limited scope of the experiments (both in terms of baselines and datasets).


**Time Spent Reviewing:**

4 hours

---

> ### Author Response · Authors · 2021-08-10
> **Thank you for your valuable comments! Response to Reviewer New7**
>
> **Significance.** We thank the reviewer for the understanding of our choices of the baselines and the datasets. We currently experiment on the two datasets studied in [30], but we will try to explore other more practical and large-scale datasets. For instance, datasets in autonomous driving that involve shifts in time and geo locations could be an option. Since our goal is to investigate if we can discover the intermediate domains, we apply the same baseline GDA algorithms as in [30]. We will try to apply other GDA algorithms on the intermediate domain sequences discovered by our algorithm in the final version.
>
> **Question for the authors.** Thanks for the questions. To investigate the variations of different trials and the effects of the number of intermediate samples, we design the following experiment. We apply IDOL (with progressive discriminator and refinement) for 10 runs with different random seeds, and record the intermediate domain index (from source = 0 to the target = M) to which each sample is assigned. For instance, if a sample is assigned to the second intermediate domain, we record 2 for this sample. After 10 runs, each sample will end up with 10 indices, and we calculate the variance. If a sample is always assigned to the same intermediate domain, the variance will be zero for this sample. We repeat the experiments with 100%/50%/25% of intermediate data (dropped randomly). The averaged variance over all samples for both the Rotated MNIST dataset (M = 21) and Portraits dataset (M = 7) are as follows:
>
> | Intermediate data  | Rotated MNIST | Portraits   |
> | :---        |    :----:   |          ---: |
> | 100%  | 1.12       | 0.147  |
> | 50%    | 1.33       | 0.150  |
> | 25%    | 1.35       | 0.136  |
>
> As shown above, the domain sequences formed in different trials are pretty consistent since the variances are small. Rotated MNIST has a higher variance than Portraits probably because the number of intermediate domains is larger. Besides, the amount of the intermediate data does not significantly affect the consistency among trials.
>
> Besides this experiment, we have also provided studies in Section 4 for further understanding of GDA and IDOL on both datasets. In Section 4.3, we studied how IDOL would perform if partial information about the intermediate domain is provided. We also studied the effect of the numbers of intermediate domains. In Section 4.4, we investigated learning with less intermediate data or adding intermediate data with noisy domain indices.
>
> **Limitations and societal impact.** Thanks for the comments. We provided some discussion in Section C and D in the supplementary material and we will further strengthen it in the final version. As far as we know, our algorithm does not imply any additional negative societal impacts beyond the existing data biases in the source, target, or intermediate domains. When applying our algorithm to discover the intermediate domains, it may unintentionally reveal the biases in the unlabeled data and one should pay attention to how to manage such information.

---

> > ### Comment · Reviewer_New7 · 2021-08-18
> > **Thanks for the response.**
> >
> > I appreciate the author's response to my comments, and I stand by my initial score.
> >
> > I really appreciate the new experiment to investigate the consistency of the domain sequences formed in different trials. I think the paper deserves to be accepted, but the main concern that stops me from increasing the score, as pointed out also by other reviewers is that the scope of the experiments in the paper are limited.

---

> > > ### Author Response · Authors · 2021-08-22
> > > **Re: Thanks for the response.**
> > >
> > > We thank the reviewer for the positive feedback about our rebuttal, and we are glad that the reviewer suggests that our paper deserves to be accepted.
> > >
> > > To respond to your concern, we think that gradual domain adaptation as a technique or scenario is widely applicable, and our proposed approach further strengthens its applicability, even when the intermediate domains are not yet separated and indexed. Our current scope of the experiments is mainly limited by the availability of benchmark datasets for gradual domain adaptation. This is mainly because GDA is a relatively new setting. Most of the existing datasets for domain adaptation are designed for the conventional unsupervised domain adaptation, in which only the source and target domain data are provided. Given (1) the superior performance of gradual domain adaptation by leveraging the additional unlabeled data and (2) the fact that real-world data change gradually more often than abruptly, we believe that more and more datasets for gradual domain adaptation will be developed and released in the near future.

---

### Official Review · Reviewer_9fF3 · 2021-07-19

**Rating:** 7
**Confidence:** 4

**Summary:**

The paper propose, IDOL, a method for gradual domain adaptation. The novelty of IDOL is that it can learn to discover the sequence of intermediate domains.

**Limitations And Societal Impact:**

The author gives a brief discussion of the limitations of this work in supplementary. Maybe the author can talk about something in more detail like when graduate domain adaptation is not applicable or when learning domain indexing is really hard. For example, in health care applications, some domain changes are caused by binary (categorical) attributes such as races, genders. Can we still apply gradual DA?

**Main Review:**

Originality:
---
This paper touches a novel problem, that is doing gradual domain adaptation with unlabeled data without pre-defined intermediate domains. The solution is technically novel. Specifically, the author leverages the idea of cycle-consistency and meta-learning to learn the domain index. I find this methodology is original and may benefit to other tasks.

Quality & Clarity:
---
For the method section, in general, the authors do a good job. However I feel there is some details about the specific design missing. I understand, at high level, the author is using a meta-learning style approach. From my understanding, meta-learning has a large computation overhead. I am curious about the algorithm complexity of the proposed method as well as experimental comparison of the learning efficiency between the fine stage algorithm against other coarse stage baselines. In appendix algorithm 3, the author uses mini-batch to update the \theta(q) for T times. I am concern the efficiency of the meta-learning when T goes large.
Another place I feel not well explained is why set q = max{q,0} instead of q = clip(q,min=0,max=1). The author says during the optimization they relax the binary vector q to be real value. I am not sure whether making a bounded value q to be unbounded will causes instability during the optimization. Can author give any theoretical verification about such relaxation?

For the experiment section, the paper delivers nice results. Table 1 shows the proposed coarse-steps are better than random indexing, while with refinement, IDOL can reach similar performance as the pre-defined index. I like the analysis in section 4.2. The visualization of per-step accuracy, relationships between learned and pre-defined indices, and class imbalance analysis provide a nice understanding of the IDOL. That being said, I still have a few question:
1. In figure 4, the author shows the target accuracy stably improves after each adaptation using IDOL sequences. Figure 4(b) shows IDOL sequences do not have much correlation with years. I am curious about what information does IDOL sequences captures.
2. The IDOL seems have worse performance than pre-defined (rotation) index in the Rotated MNIST dataset. I am assuming in this task, rotation is the best domain index. Can authors analysis how close the learned IDOL align with the rotation? If IDOL is not perfectly align with rotations, is there any potential reasons that hinders it learn the best domain index?

Significance:
---
The paper propose a new problem of doing GDA with undefined intermediate domains. The problem is important. The result presented is significant and could be a nice baseline for the future research.

In summary, I am voting for the acceptance due to its novelty and nice experiments. My score is not final and will depend on how the author address my questions.

**Time Spent Reviewing:**

10

---

> ### Author Response · Authors · 2021-08-10
> **Thank you for your valuable comments! Response to Reviewer 9fF3**
>
> We thank the reviewer for the valuable comments.
>
> **Computation cost.** We provide an estimation of the wall-clock time of the coarse stage and fine stage in the **general response** above. IDOL provides several choices of the coarse stage for its users with different computations budgets. For learning the data parameter $q$ using meta-learning, it requires back-propagating the gradients of $\theta$, which approximately takes three-time of the computation of a standard forward-backward pass. In Algorithm 3, we found that a small number of steps T is sufficient for the forward and backward model adaptation and we set T to be 10 in all experiments as mentioned in Line 61 in the supplementary material.
>
> **On learning $q$.** The way we relax the binary-valued $q$ by real values is related to the linear programming relaxation of an integer programming problem. It has been widely applied, for example, in discovering sub-domains within a dataset [14]. Theoretically, we should constraint each element of $q$ to be within $[0, 1]$. Empirically, we found that even without the clipping operation to upper-bound q (i.e., $\max{q, 0}$), the values in q do not explode and the algorithm is quite stable. We did try the clipping way as mentioned by the reviewer, and we see negligible difference in the resulting accuracy of GDA. Another reason we choose to use $\max{q, 0}$ is to be consistent with the common practice of using meta-reweighting [26, 50]. We thank the reviewer for the comment and we will clarify this design choice and provide more discussion in the final version of the paper.
>
> **Experiments (question 1).** Thank you for the question. In Figure 4 (b), although the IDOL sequence does not perfectly align with years, we still find a reasonable 0.727 correlation. We note that in the Portraits dataset [12], the domain changes between the source and target domains result from several factors such as fashions, hairstyles, eyeglasses, and lip curvatures that may not perfectly align with years (please see [12] for discussions). Even within the same year, individuals could have variations in these factors as well. We hypothesize that IDOL can discover the domain sequence that reflects a smoother transition of these factors than years.
>
> **Experiments (question 2).** We would like to note that, while on the Rotated MNIST dataset, IDOL based on the progressive domain discriminator does not outperform the pre-defined rotation sequence (87.5 vs. 87.9), IDOL based on the pre-defined sequence can outperform the rotation sequence (93.3 vs. 87.9). In other words, the rotation index given by the dataset may not be the best domain index. We attribute this to how the Rotated MNIST dataset is created. Concretely, the dataset treats the original MNIST data as 0-degree rotation, and artificially rotates the data given a rotation index. However, for data in the original MINIST dataset, there may already exist variations in terms of rotations. This can be seen in Figure 3: in the first row of 0-degree rotation based on the pre-defined index, the examples of digit 1 do have slight rotations. In this situation, we believe that IDOL could better capture the true rotation of each example. As suggested, we compute the correlation between the IDOL sequence based on the progressive domain discriminator and the pre-defined sequence and it is 0.807. The correlation between the IDOL sequence based on the pre-defined sequence and the pre-defined sequence is 0.821. The latter indicates that IDOL does refine the pre-defined sequence to achieve a better GDA performance.
>
> **Limitations.** Thank you for the comments and we will provide more discussions in the final version. GDA will be less effective if the additional unlabeled data besides the source and target domains do not gradually bridge the two domains or contain outliers. We investigated these briefly in Section 4.3 (Line 298-300) and Section 4.5. If there are unlabeled data close to the source or target domains but are not distributed in between of them (e.g., if the source/target are digits with 60/0-degree rotation, then digits with 70-degree are one of such cases), IDOL may still include those data into the intermediate domains and could potentially degrade the overall performance. We thank the reviewers for bringing up the health care applications. In the cases that the domain changes are caused by the distributions of the categorical attributes, for example, the source data are from 70% males and 30% females while the target data are from 30% males and 70% females, GDA should still be applicable if we have unlabeled data resulted from intermediate proportions of the two domains.

---

> > ### Comment · Reviewer_9fF3 · 2021-08-23
> > **Good job**
> >
> > I appreciate the author's effort in the rebuttal. All my questions are addressed well.
> >
> > I really like the author's explanation about the comparison between IDOL sequences and the predefined sequences. In both cases, Portrait and Rotate-MNIST, I find the author's explanations are convincing. I did not think that original MNIST digits may have different rotations. It is a very nice point. I do learn something from the author's rebuttal.
> >
> > In all, I would like to congratulate the author for doing a great job of both the submission and the rebuttal.

---

### Author Response · Authors · 2021-08-10
**General response including computation cost**

We thank the reviewers for their valuable comments. We are glad that the reviewers found that our paper touches on a “novel” and “important” problem (R1); our methodology is “novel” (R1, R2), “original” (R1), and “theoretically motivated” (R2). All reviewers acknowledge that the performance of IDOL is comparable to pre-defined indexes. We first answer the general question on the computation cost, followed by separately responding to the comments by each reviewer. We will incorporate all the feedback in the final version.

R1: Reviewer 9fF3
R2: Reviewer New7
R3: Reviewer B4p7


**Computation cost.** We provide more information about the computation cost and present the execution time. For coarse scores, the confidence is simply by applying the classifier to the input data. For the Portraits dataset, calculating the manifold distance via [45] takes about 20 seconds with Intel i9-9960X CPU. The domain discriminator (both w/ or w/o progressive training) and the fine stage involve GPU computation. Using a GeForce RTX 2080 Ti GPU, training a domain discriminator takes about 4 seconds. Progressive training takes about $M$ times longer, where $M$ is the number of intermediate domains (which is 7 here). This is because it trains the domain discriminator for $M$ rounds. For the refinement stage proposed in Section 3.4 based on meta-learning, the inner loop's forward/backward adaptation (Line 221 & 223 and Algorithm 3 in the supplementary material) is trained for T = 10 mini-batches (with size 128), respectively. For the outer loops, we trained for 30 epochs in total. Discovering one intermediate domain takes about 44 seconds and we perform it for $M$ rounds.

---

### Decision · Program_Chairs · 2021-09-27

**Decision:**

Accept (Poster)

**Comment:**

The paper considers gradual domain adaptation without access to intermediate sequence of distributions between source and target. This is a relatively new problem and a very important one. Author discussions helped address the reviewer concerns and therefore I suggest the paper to be accepted. I ask the authors to please include the additional information discussed in the rebuttal period in the camera ready.  I also strongly recommend the authors add more explanations about the limitations of the current experiment setting (GDA) and discuss the possible ways to actively address these issues. Although the paper is interesting and the results are a good first step to tackle the problem of gradual domain adaptation, the experiment are limited, limitations of the proposed method are not investigated and the baselines are simple. Therefore, the paper does not qualify for a spotlight.